# Improving the Yield and Quality of *Morchella* spp. Using Agricultural Waste

**DOI:** 10.3390/jof11100703

**Published:** 2025-09-28

**Authors:** Jiawen Wang, Weiming Cai, Qunli Jin, Lijun Fan, Zier Guo, Weilin Feng

**Affiliations:** 1College of Horticulture, Zhejiang Agriculture and Forestry University, Hangzhou 311300, China; wangjw@zaas.ac.cn; 2Zhejiang Key Laboratory of Biological Breeding and Exploitation of Edible and Medicinal Mushrooms, Institute of Horticulture, Zhejiang Academy of Agricultural Sciences, Hangzhou 310021, China; jinql@zaas.ac.cn (Q.J.); fanlj@zaas.ac.cn (L.F.); guoze@zaas.ac.cn (Z.G.)

**Keywords:** *Morchella* spp., tomato substrate, mushroom residues, coconut shells, bacterial community, fungal community, yield, quality

## Abstract

*Morchella* spp. is a type of valuable and rare edible fungi cultivated in soil. Optimization of the cultivation medium for *Morchella* spp. is key to obtaining high-efficiency production in an ecologically friendly manner. Recently, the sustainable resource utilization of agricultural waste has gathered attention. Specifically, reusing tomato substrate, mushroom residues, and coconut shells can lower the production costs and reduce environmental pollution, demonstrating remarkable ecological and economic benefits. To determine the soil microbial communities of *Morchella* spp. using different culture medias and influencing factors, this study analysed the relative abundance of bacterial and fungal communities in natural soil, soil with 5% tomato substrate, soil with 5% mushroom residues, and soil with 5% coconut shells using Illumina NovaSeq high-throughput sequencing. In addition, intergroup differences, soil physiochemical properties, and product quality were also determined. Results demonstrated that agricultural waste consisting of mushroom residues, waste tomato substrate, and coconut shells can improve the efficiency of *Morchella* spp. cultivation. When considering yield and quality, mushroom residue achieved the highest yield (soil nutrient enrichment), followed by tomato substrate (water holding + grass carbon nutrient). All three types of agricultural waste promoted early fruiting, significantly increased polysaccharide, crude protein, and potassium content, and lowered crude fat and fibre. In regard to soil improvement, the addition of different materials optimized the soil’s physical structure (reducing volume weight and increasing water holding capacity) and chemical properties (enrichment of nitrogen, phosphorus, and potassium, regulating nitrogen and medium trace elements). For microbial regulation, the added materials significantly increased the abundance of beneficial bacteria (e.g., Actinomycetota, Gemmatimonadota and *Devosia*) and strengthened nitrogen’s fixation/nitration/decomposition functions. In the mushroom residue group, the abundance of *Bacillaceae* was positively related to yield. Moreover, it inhibited pathogenic fungi like *Mortierella* and *Trichoderma*, and lowered fungal diversity to decrease ecological competition. In summary, mushroom residues have nutrient releasing and microbial regulation advantages, while tomato substrate and coconut shells are new high-efficiency resources. These increase yield through the “physiochemical–microorganism” collaborative path. Future applications may include regulating the function of microorganisms and optimizing waste preprocessing technologies to achieve sustainability.

## 1. Introduction

*Morchella* spp. belongs to *Morchella*, Moechella-ceae, Pezizales, Asco-mycota [1], and is a type of rare edible and medicinal fungus. Its fruiting body is known to taste delicious and is also known for its medicinal properties, including anti-oxidative, immunoregulatory, anti-inflammatory, anti-tumoral, and anti-obesity characteristics [2]. With recent breakthroughs in exogenous nutrient bags, the artificial cultivation of *Morchella* spp. has rapidly expanded in China, as evidenced by an increasing cultivation scale and yield each year [3]. However, cropping obstacles caused by intensive planting have become increasingly prominent, which mainly manifest as poor hypha growth, no primordium differentiation, decreased yield, decreased size of the ascocarp, and serious diseases and insects. These can lead to reduced yields and in some instances total crop failure [4]. *Morchella* spp. is known as a strict type of heterotrophic fungus. The development of its hypha growth and fruiting body highly relies on the physiochemical properties of soil microorganisms and the dynamic balance of microbial communities. Hence, optimizing the culture medium is essential to overcome cropping obstacles and improve sustainability.

Agricultural waste is known to be low cost, with extensive uses, and good physiochemical properties. As an additive to optimize the cultivation medium of *Morchella* spp., agricultural waste can increase economic benefits and relieve environmental pollution. Based on previous studies and current resources, three types of common agricultural waste, including tomato substrate, mushroom residues, and coconut shells, were selected for this study.

Recently, soilless culture has been extensively used in facility-based tomato production [5]. With an increase in facility-based tomato planting in China, tremendous amounts of tomato substrate waste are produced during production. As a result, this leads to the accumulation of nitrogen, phosphorus, potassium, and base ions [6]. Hence, tomato substrate waste is yet to be applied to cropping, including studies investigating substrate improvement for *Morchella* spp. At present, composite substrate is the most common substrate for tomato cultures. Traditional composite substrate is mainly composed of a mixture of vermiculite and grass carbon at different mixing ratios. Vermiculite can optimize physical properties like porosity, whereas grass carbon provides organic substrate nutrients and can meet the requirements of a rhizospheric environment for crop growth and development [7]. As a result, the accurate processing and recycling of waste tomato substrate can save resources, lower the extra consumption of waste substrate during processing, and improve the nutritional value and benefit of economic crops.

Mushroom residues are substrate remains after the harvest of edible mushrooms. It is estimated that the annual yield of mushroom residues in China exceeds 100 million tons, which results in tremendous environmental pressure and requires processing costs. Mushroom residues have loose and porous structures. Some studies have reported that mushroom residues contain nutrients which are necessary for the growth and reproduction of multiple edible mushrooms, such as crude protein, crude fat, crude fibre, crude polysaccharide, trace elements, vitamins, and minerals [8]. It is relatively easy to recycle mushroom residues from edible fungi. When developed into new culture substrate for edible fungi, mushroom residues can lower costs effectively. Wang Shangrong et al. found abundant *Morchella* spp. in fast-growing poplar forest land after the corrosion of oyster mushroom and Coprinus comatus residues from subsurface cultivation [9]. This phenomenon revealed that mushroom residues might provide an appropriate substrate for the growth of *Morchella* spp. Further research confirmed these beneficial effects of mushroom residues. Chen Xiheng found that optimal yields of *Morchella* spp. can be achieved using 60% of farmland soil, 20% of mushroom residues, 10% of perlite, and 10% of burnt earth [10]. Song Wei et al. highlighted that the addition of Shiitake mushroom residues and organic fertilizers improved soil quality and promoted the growth of *Morchella* spp. [11]. With regards to the mechanism of action, Shen Tong et al. found that the porous structures of mushroom residues could relieve substrate hardening and promote oxygen intake of the hypha from *Morchella* spp., thus enabling improved hypha activity and differentiation efficiency of fruiting bodies [12]. In summary, mushroom residues have multiple functions in the improvement of cultivation substrate, including promoting hypha growth and increasing yields. Hence, mushroom residues should be used to facilitate resource recycling.

In the cultivation substrate of *Morchella* spp., the recycling of coconut shells has attracted increasing attention in recent years. Due to the loose and porous fibre structures, strong water holding and air permeability characteristics, as well as high cellulose, lignin, and nutrient content (e.g., Ca^2+^), coconut shells are similar to mushroom residues in term of physical properties and nutrient release. Additionally, from the perspective of microbial regulation, the lignocellulose of coconut shells might stimulate flora colonization and interactions during decomposition. The Kunming Institute of Botany, Chinese Academy of Sciences, found that bacteria participating in nitrogen fixation (*Arthrobacter* and *Bradyhizobium*) and nitration (e.g., *Nitrospira*) predominated in soils with high yields of *Morchella* spp. (yield ≥ 1500 kg/ha) [13]. Carbon sources released from the degradation of coconut shells in soil may provide metabolism substrate for these bacteria, thus strengthening the transformation of nitrogen in soil. A research team led by Zhang Chen proved that microorganisms in the hypha phase of *Morchella* spp. were mainly dominated by carbohydrate metabolism [14]. Moreover, the gradual decomposition of fibre in coconut shells can provide carbon sources for microorganisms like Proteobacteria and Gemmatimonadetes, accelerating substrate degradation and releasing minerals [15]. In addition, Zhang Xiangfeng et al. studied the microorganisms of wild *Morchella* spp. and found that the microbial diversity of surface soils was significantly higher than that of deep soils due to rich organic matter and loose structures [16]. Coconut shell substrate can strengthen microbial functional diversity and ecological stability by simulating similar physical environments. These studies have systematically disclosed the scientific basis and feasibility of using coconut shells as a culture substrate additive for *Morchella* spp. The scaled application of coconut shells can relieve pressure, lower industrialized production loss, promote the recycling of agricultural waste, and improve the industrial production of edible mushrooms.

The hypha growth and fruiting body development of *Morchella* spp. relies highly on the soil microenvironment. Studies have reported that soil microorganisms play dual roles in vital activities of *Morchella* spp. On the one hand, the rich bacterial communities in soils are very favourable for the growth of *Morchella* spp. [17]. Research on the microflora in soils and fruiting bodies of *Morchella* sextelata under greenhouse culture showed that *Pseudomonas* can promote nutrient and reproductive growth of *Morchella* spp. [18]. On the other hand, fungal (e.g., canker and white mould disease) and bacterial diseases (e.g., soft rod) in successive cropping soils have become key obstacles in sustainable industry development [19]. Although previous studies have proved that structural changes of microflora are major causes of successive cropping obstacles, current studies mainly focus on single microbial groups (e.g., fungi or bacteria). There is a lack of systematic studies, including those investigating the regulatory effect of exogenous additives on the interaction network of flora.

Recycling of agricultural waste opens a new avenue in the improvement of soil microenvironments. As a type of effective soil improvement agent, organic materials like tomato substrate, mushroom residues, and coconut shells can directly affect crop yield and quality by changing soil pH, porosity, soil nutrients, and other physiochemical properties [10,11,12], as well as increase nutrient cycling efficiency by reconstructing the microflora structure, thus facilitating the growth and development of crops [15]. The unique structure of coconut shell fibres might offer an ideal substrate for hypha formation in *Morchella* spp., while the lignin degradation flora in mushroom residues might participate in its metabolism. However, research on the application of agricultural waste to enhance *Morchella* spp. production is lacking. Few studies on the application of mushroom residues have been reported, and there is no systematic study on the regulatory mechanism of waste tomato substrate and coconut shells. Namely, effects on the yield and quality of fruiting bodies from multiple perspectives of “soil physiochemical properties-microbial interaction-host responses”.

Based on the above scientific problems, this study sought to analyse the microbial interaction network systems of soil after the addition of tomato substrate, mushroom residues, and coconut shells. Here, Illumina NovaSeq PE30 high-throughput sequencing based on the innovative integration of multimodal research methods was used. Moreover, physiochemical indicators like soil pH, nitrogen, phosphorous, and potassium were monitored to explore the cascade effect of “waste adding—soil improvement—microbial regulation—quality & yield”. The research results can offer new approaches to improve the culture substrate for *Morchella* spp. based on the regulation of microbial communities, thus transforming the edible mushroom industry using resource recycling.

## 2. Materials and Methods

### 2.1. Experimental Design and Sample Collection

A field test was carried out in the Yangdu Base of Zhejiang Academy of Agricultural Sciences in Haining City, Zhejiang Province, East China (30.5142256° N, 120.6759733° E, elevation = 3.7 m). *Morchella* spp. was planted in boxes (40 cm (length), 30 cm (width), 25 cm (height)) on 19 November 2024. All boxes were cleaned and sterilized using saturated bleaching powder in advance. Cultivation soils were collected from sandy loams with good air and water permeability in Lanxi City, Zhejiang Province. Boxes were supplemented with either 5% tomato substrate (waste substrate after soilless tomato culture, vermiculite: grass carbon = 3:1, leaching for 24 h before the use), 5% mushroom residues (waste mushroom residues after oyster mushroom culture), or 5% coconut shells (immersed in water for 24 h before use). These boxes were placed in a unified environment, along with a Lanxi natural soil box as a control. The soil height was determined at 20 cm, and the local temperature, air humidity, and soil humidity were 27 °C, 70%, and 80%, respectively. The temperature, air humidity and soil humidity were detected in real time during the whole cultivation process to ensure that each sample was the same and within the most suitable growth range of Morchella. Each sample had three replicates, which were labelled as A3, B3 and C3 and control (Figure 1).

Fruiting bodies were harvested twice according to ripening times, on 18 February 2025, and 25 February 2025. Soil samples were collected in the late ripening phase of fruiting bodies, on 25 February 2025, which corresponds to 98 days after sowing. The cultivation soil without waste (T1), with tomato substrate (A3), with mushroom residues (B3), and with coconut shells (C3) were collected. A five-point sampling method was adopted. In each box, five sampling points were chosen randomly and surface soil was removed using a sterilized shovel. Soil at a depth of 3–5 cm was collected, mixed evenly, and then divided into two parts. The first part (approximately 300 g) was used to test soil physiochemical properties. The second part was sieved (2 mm-sized sieve) to remove large particles. Subsequently, 50 g of sample was rapidly placed into a sterile tube, numbered, and stored in the dry ice bucket. The collected soil samples were transported to the laboratory as soon as possible, and the sterilized soil samples were stored in a −80 °C refrigerator for DNA extraction and sequencing. Samples were transported in liquid nitrogen to the Shanghai Meiji Biotechnology Information Technology Co., Ltd. (Shanghai, China) for Nextseq PE300 sequencing. To reduce experimental errors, soil physiochemical properties and high-throughput sequencing included six biological repetitions. The total sample size for sequencing was 24. We combined the calculations of mean and variance.

### 2.2. Agronomic Characteristics Test of Fruiting Bodies

Images of fruiting bodies were captured after harvest to observe their form. These were weighed and their yields calculated. The nutrient element indicators of *Morchella* spp. were tested according to national standards (NY/T) [20]. Polysaccharide content was tested using anthrone-sulphuric acid colorimetry. Crude protein content was tested using the Kjeldahl method. After sulphuric acid–hydrogen peroxide boiling, the total potassium (TP) content was tested using the atomic absorption flame photometer method. The crude fat content was measured using the Soxhlet extraction method, and the cellulose content was tested using the gravimetric method.

### 2.3. Test of Physiochemical Properties of Cultivation Soils

Soil chemical indicators are referred to as NY/T, LY/T. Total nitrogen content in soils were tested using the element analyser. Soil pH and available phosphorus (AP) were tested using the NaHCO_3_ digestion–molybdenum–sulphur antimony colorimetric method. Available potassium (AK) content was tested using the ammonium acetate solution extraction-flame photometer method. Ammonium nitrogen was tested using the potassium chloride solution extraction–indophenol blue colorimetric method. Nitrate nitrogen was tested using the potassium chloride solution extraction–dual-wavelength colorimetric method. Exchangeable calcium magnesium ion content was tested using the ammonium acetate exchange method. Available zinc was tested using the DTPA–TEA extraction method [20,21]. The determination results of all physical and chemical properties were repeated six times, the average value was taken for statistical analysis, and the significant difference analysis was carried out by excel table.

### 2.4. Microbial Diversity Test

The FastDNA SPIN Kit (MP Biomedicals, Irvine, CA, USA) for Soil was used to extract soil DNA. After extraction, DNA segment size was tested using the agarose gel electrophoresis method. DNA concentration and purity were tested using the NanoDropOne (Thermo Fisher, Waltham, MA, USA). All samples that met quality control requirement were sent to Shanghai Meiji Biotechnology Information Technology Co., Ltd. for Nextseq PE300 sequencing. The primers and sequences used are listed in Table 1. PCR conditions included: 3 min pre-degeneration at 95 °C, 30 s degeneration at 95 °C, 30 s renaturation at 55 °C, 45 s extension at 72 °C, 35 cycles; 10 min extension at 72 °C.

### 2.5. Data Analysis and Visualization

In this study, pair-end (PE) 300 bp sequencing was carried out on the Illumina Nextseq PE300 sequencing platform (Illumina, San Diego, CA, USA). After data acquisition, samples were divided for quality control and filtering purposes according to the quality of PE Reads. Next, samples were pooled, and high-quality sequences were generated. The data was then optimised through sequence denoising (using the DADA2 or Deblur method) to obtain the Amplicon Sequence Variant (ASV). Species classification, flora diversity, species differences, material correlation, and functions were analysed and predicted based on expression sequences and abundance information. Finally, microbial community structures and functional characteristics were obtained through multivariate statistics and visualization methods. We used QIIME2 version 2024 with the DADA2 denoising method. For functional prediction, we employed PICRUSt2 version 2.2.0 and FUNGuild version 1.0. The sequencing data was uploaded onto the National Centre for Biotechnology Information (nih.gov), with the sequence No. PRJNA1277449 (16S) and No. PRJNA1280193 (ITS). Furthermore, a joint analysis on species composition, functions, and relationship with soil physiochemical properties was carried out.

## 3. Results

### 3.1. Effects of Different Additives on the Yield and Quality of Morchella spp.

Variations in agronomic characteristics of *Morchella* spp. under different culture substrates are shown in Figure 2. The yields and ripening time of *Morchella* spp. varied significantly among different additives. Results showed that the control group (T1) ripened on 25 February 2025, with a small yield. On the other hand, all treatment groups with different substrates ripened earlier and achieved higher yields than the control group. The coconut shells group ripened the earliest, followed by the tomato substrate group. The mushroom residue group achieved the highest yield. Fruiting bodies were collected on 18 February 2025, and 25 February 2025. All samples were weighed, and total yields were calculated (Figure 2). The yields of A3 (5266.23 kg/hm^2^), B3 (5847.22 kg/hm^2^), and C3 (4210.28 kg/hm^2^) were all significantly higher by 259.95%, 299.66%, and 187.78% when compared to that of T1 (1463.06 kg/hm^2^).

Table 2 shows that polysaccharides, crude protein, TP (total potassium ion), crude fat, crude fibre, and water content changed considerably with additives. Specifically, polysaccharide and TK content increased significantly under all substrates. Polysaccharides were increased the most in the substrate with mushroom residues, while TK were increased the most in the substrate with coconut shells. Crude fat and crude fibre content declined significantly compared to those of the control group.

### 3.2. Effects of Substrate Types on Soil Physiochemical Properties for Morchella spp. Culture

In Table 3, indicators of all samples were significantly different from those of the control group except for pH, nitrate nitrogen, and available zinc in soils with tomato substrate. Water content, pH, TK, nitrate nitrogen, and exchangeable magnesium of cultivation soils with different additives all increased to some extent compared to those of Lanxi natural soil, while ammonium nitrogen declined. Specifically, the highest water content was observed in soil with tomato substrate, while the highest TN (total nitrogen), AP (rapidly available phosphorus), AK (rapidly available potassium), nitrate nitrogen, exchangeable magnesium, and available zinc content were achieved in soils with mushroom residues. It is worth noting that the ammonium nitrogen content of all experimental groups was lower than that of the control group, while the nitrate nitrogen content was higher.

### 3.3. Bacterial Community Analysis in Cultivation Soils with Different Additives

#### 3.3.1. Bacterial Diversity in Cultivation Soils

PCoA (Principal Coordinates Analysis) analyses on cultivation soils of *Morchella* spp. with different additives were carried out, resulting in the data shown in Figure 3. The differences in bacterial structures and the diversity of samples increased with the increase in their distances. According to the results, the contribution rates of the effects of bacterial community structures were expressed by the PC1 (Main coordinate 1) and PC2 (Main coordinate 2) axes, at 72.15% and 5.4%, respectively. The total degree of explanation reached 77.55%. The bacterial structure of the coconut shells group was the most similar to that of the control group. Soils with tomato substrate, mushroom residues, and coconut shells were in different quadrants, indicating that the bacterial structures and diversity in cultivation soils for *Morchella* spp. changed significantly with the use of different additives.

#### 3.3.2. Species Composition Analyses on the Phylum and Genus Levels in Cultivation Soils

To show bacterial community characteristics in cultivation soils with different additives for *Morchella* spp. comprehensively, species composition on the phylum and genus levels were analysed. It can be seen from Figure 4A that on a phylum level, Pseudomonadota, Bacteroidota, Chloroflexota, Bacillota, and Actinomycetota were the dominant bacteria. Significant differences in phylum composition were observed in cultivation soils with different additives, especially in soils with mushroom residues. It is important to note that the abundance of Actinomycetota in the tomato substrate group was 8.81%, which was 74.11% higher than that of the control group. The abundance of Actinomycetota and Gemmatimonadota in the mushroom residue group were 12.77% and 5.86%, which were increased by 152.37% and 53.81% compared to that of the control group, respectively. The abundance of Gemmatimonadota in the coconut shells group was 3.84%, which was increased by 0.79% than that of the control group. The species composition on the genus level is shown in Figure 4B. The relative abundance of *Devosia* in the tomato substrate group and coconut shells group were 3.37% and 1.33%, an increase of 185.59% and 12.71% compared to that of the control group, respectively. Additionally, the abundance of *Bacillaceae* in all experimental groups increased significantly (1.11%, 3.37%, and 0.61%), which were increased by 4, 13, and 2 times compared to that of the control group (0.25%), respectively.

#### 3.3.3. Analysis of Bacterial Community Differences Among Cultivation Soils with Different Additives

To evaluate differences in the effects of different treatments on abundance of components, a LEfSe (Linear discriminant analysis Effect Size) based on linear discriminant analysis (LDA) was used. Firstly, LDA was plotted to assess the bar chart, aiming to recognize bacteria with significant differences among different groups (significance threshold set to LDA = 4). Higher LDA values indicate greater effects of soil treatments on community species abundance. It can be seen from Figure 5A that the relative abundances of a: Acidobacteriota in the control group, g1: Hyphomicrobiales in the tomato substrate group, c: Bacillota in the mushroom residue group, and e: Bdellovibrionota in the coconut shells group were relatively high.

*t*-test results are shown in Figure 5B. In the control group, Bacteroidia and Bacteroidota were major bacterial classes with significant differences compared to other species. In soils with tomato substrate, Alphaproteobacteria was significantly different within groups. Bacillota was the main bacterium that caused significant differences between soil samples between mushroom residues and other experimental groups. In soils with coconut shells, Pseudomonadota and Gammaproteobacteria resulted in significant differences, and their degrees of influence all reached 5.

#### 3.3.4. Prediction of Potential Functions of Bacteria in Cultivation Soils with Different Additives

It can be seen from Figure 6 that the functions of bacteria in all samples were predicted using the PICRUSt2 method based on the COG (Clusters of Orthologous Groups) database. Results showed that metabolism (34.23–35.51%) was the major pathway of bacteria in the cultivation soil of *Morchella* spp., with the major metabolism pathways being amino acid transport and metabolism (9.74–10.50%), and inorganic ion transport and metabolism (5.93–6.26%). The metabolism level of cultivation soil with agricultural waste was slightly higher than that of natural soil.

#### 3.3.5. Correlation Analysis Between Yield and Quality of Fruiting Bodies and Key Bacteria

A joint analysis was carried out because the nutrient quality and yield improvement of *Morchella* spp. may correlate with the soil’s physiochemical properties. Results are shown in Figure 7. According to a community composition analysis, all physiochemical indicators, the top six bacteriophyta (Pseudomonadota, Bacteroidota, Chloroflexota, Bacillota, Actinomycetota and Gemmatimonadota), and two critical genera (*Devosia* and *Bacillaceae*) were chosen for analysis. Results showed that TN, AP, AK, nitrate nitrogen, exchangeable magnesium, and available zinc showed significantly positive correlations with Chloroflexota, Bacillota, and *Bacillaceae*, but significantly negative correlations with Pseudomonadota and Bacteroidota. The yield was positively correlated with Chloroflexota, Gemmatimonadota, and *Devosia*, and significantly positively correlated with TN, AP, AK, nitrate nitrogen, exchangeable magnesium, Actinomycetota, Bacillota, and *Bacillaceae*. The yield was significantly negatively correlated with ammonium nitrogen. The quality showed a significant negative correlation with Actinomycetota.

### 3.4. Fungi Community Analysis in Cultivation Soils with Different Additives

#### 3.4.1. Fungal Diversity of Cultivation Soils

Natural soil, soil with tomato substrate, mushroom residues, and coconut shells were chosen for Venn diagram analysis. Natural soil, soils with tomato substrate, mushroom residues, and coconut shells contained 235, 202, 192, and 213 OTUs (Operational Taxonomic Units), respectively. There were 69 special OTUs in natural soil, 44 in soil with tomato substrate, 43 in soil with mushroom residues, and 55 in soil with coconut shells. There were 79 common OTUs among the above four samples. It can be seen from Figure 8 that natural soil contained the highest number of OTUs, followed by soil with coconut shells, tomato substrate, and mushroom residues, successively. This revealed that the number of fungi communities in natural soil was the highest, but this decreased to different extents after the addition of different agricultural waste. The number of fungi communities in soil with mushroom residues was the lowest.

#### 3.4.2. Species Composition Analyses at a Fungal Phylum and Genus Level in Cultivation Soils

To comprehensively show fungal community characteristics in cultivation soil with different additives, species composition at a phylum and genus level were analysed. According to a material analysis at a phylum level, the dominant fungi were Ascomycota and Mortierellomycota. The relative abundance of Mortierellomycota declined compared to that of the control group. On a genus level, the dominant fungi were *Morchella*, *Furcasterigmium*, and *Mortierella*. The relative abundance of *Furcasterigmium* increased compared to that of the control group, while the relative abundance of *Morchella* and *Mortierella* decreased to different extents. Specifically, the relative abundance of *Mortierella* was 12.63%, and it decreased from 49.17%, 16.71%, and 42.20% to 6.42%, 10.51%, and 7.30% in A3, B3, and C3, respectively. Additionally, the relative abundance of *Trichoderma* was 0.91% in the control group, and it decreased significantly to 0.10%, 0.77%, and 0.09% in A3, B3, and C3, respectively. Results are shown in Figure 9A,B.

#### 3.4.3. Analysis of Fungal Community Differences Among Cultivation Soils with Different Additives

A ring-shaped tree graph was obtained from the LEfSe analysis of inter-group differences in fungal communities. Circles from the inside to the outside represent species classes. It can be seen from Figure 10A that the relative abundance of c: Mortierellomycota in natural soil, d: Rozellomycota in soil with tomato substrate, f: Eurotiomycetes in soil with mushroom residues, and a: Basidiomycota in soil with coconut shells were relatively high.

The *t*-test results are shown in Figure 10B. In the control group, Hypocreales was the major fungal category that caused significant differences when compared to other experimental groups. In soil with tomato substrate, Cordycipitaceae significantly affected inter-group differences. In soil with mushroom residues, the degree of influence of Plectosphaerellaceae, Glomerellales, and *Furcasterigmium* on inter-group differences significantly reached 5. *Lecanicillium* was the major fungus that caused significant differences between soil samples with coconut shells and other experimental groups.

#### 3.4.4. Prediction of Potential Functions of Fungi in Cultivation Soil with Different Additives

To further determine functional microbial states of fungal communities in cultivation soil with different additives, data with a confidence of “Highly Probable” and “Probable” were chosen to analyse the functions of soil fungi in different groups. Data reliability through confidence screening was assured and function annotation was performed using FUNGuild. Combined with abundance statistics and inter-group comparisons, the effects of agricultural waste on fungal functional communities in soil, particularly the dynamic changes of saprophytic and pathogenic microorganisms, were revealed. It can be seen from Figure 11 that saprophytic fungi took a dominant role in all soil samples, while the proportion of pathogenic fungi was relatively low.

#### 3.4.5. Correlation Analysis Between Yield and Quality of Fruiting Bodies and Key Fungi

According to an analysis of fungal community differences, all physiochemical indicators, the top two fungal phyla (Ascomycota and Mortierellomycota), and the four key genera (*Morchella*, *Furcasterigmium*, *Mortierella*, and *Trichoderma*) were chosen for the combined analysis. Results are shown in Figure 12. Evidently, *Furcasterigmium* had significant correlations with physiochemical indicators, whereas the remaining fungi were weakly correlated with physiochemical indicators. The total correlation indicators were lower than those of bacterial communities. The yield was significantly positively correlated with *Furcasterigmium*, but negatively correlated with Mortierellomycota, *Morchella*, *Mortierella*, and *Trichoderma*.

## 4. Discussion

### 4.1. Effects of Agricultural Waste on the Yield and Quality of Morchella spp.

The addition of agricultural waste can significantly influence the yield and quality of *Morchella* spp. Guo Ying et al. found that adding mushroom residues increased the yield of needle mushrooms [22]. He Ruifeng et al. discovered that mushroom residues could be used to cultivate Agaricus bisporus and promote fruiting earlier than traditional faeces and grass [23]. These studies are in concordance with experimental results obtained in this study. This study also proved that tomato substrate and coconut shells can promote earlier fruiting and increase yield. With respect to quality, polysaccharides, crude proteins, TK, crude fat, crude fibre, and water content of different groups were significantly influenced by waste additives. The mineral element enrichment of *Morchella* spp. in different growth environments is sensitive to multiple biological and non-biological factors [24], thus resulting in different contents of elements among different groups. In all experimental groups, polysaccharide content increased significantly. Adding agricultural waste was similar to straw turnover, which can increase utilization of nitrogen fertilizer and activity of the carbon pool. *Morchella* spp. can acquire carbon sources by decomposing carbon-containing organic matter and facilitating synthesis of polysaccharides [25]. In particular, the polysaccharide content of fruiting bodies increased the most after adding mushroom residues. This is in agreement with previous studies on Auricularia auricula-judae cultivation in soil with Ganoderma lucidum residues [26]. Potassium content increased significantly after adding agriculture waste, particularly coconut shells. The Chinese Academy of Agricultural Sciences argued that potassium accounts for 40–45% of mineral elements in edible mushrooms, and that these are the ideal K supplement due to their low sodium and low fat content [27]. Crude fat and crude fibre content decrease significantly after agricultural waste was added. Edible mushrooms with higher content of polysaccharides, crude protein, and crude fat had better quality. However, consumers concerned about obesity prefer foods with low fat content. Edible mushrooms with lower content of crude fat and crude fibre have better quality [28]. The effect of adding agricultural waste to improve nutrient quality of *Morchella* spp. agreed with this general trend.

### 4.2. Effects of Agricultural Waste on the Physiochemical Properties of Cultivation Soil

Adding agricultural waste can significantly affect the physiochemical properties of cultivation soils for *Morchella* spp. Since pH and EC (Electrical Conductivity) in substrate increase after tomato planting, nitrogen, phosphorus, potassium, and base ions accumulate in the soil [6]. Tomato waste substrate is hardly used in crop cultivation and no research on its contribution to *Morchella* spp. cultivation has been reported to date. In this study, tomato substrate was leached for 24 h in the study area. The pH and EC values of the tomato substrate before impregnation were 6.2 and 4.5 mS/cm, respectively. After impregnation, the pH increased to 6.8 and the EC value decreased to 1.6 mS/cm. Related reports showed that soil pH 6.5~7.5 was the best for *Morchella* cultivation [29]. Although there is no clear report on the optimal EC value for *Morchella* cultivation, it is generally believed that excessive salt leads to ion accumulation, which may inhibit the growth of mycelium. Therefore, the decrease in EC value is usually beneficial to the growth and development of edible fungi. After tomato substrate was added, the soil’s physiochemical properties, including pH, nitrogen, phosphorus, potassium, and base ions, were tested within normal ranges. Vermiculite and grass carbon are major components of tomato substrate. Vermiculite can optimize physical properties like porosity. In this study, the tomato substrate group had the highest water content in soil. *Morchella* spp. has a high requirement for water during growth. An elevated water-holding capacity is thus conducive to increasing the yield of *Morchella* spp. [30]. Grass carbon has relatively high nutrient content. Tomato substrate can improve soil physiochemical properties to some extent, leading to relatively ideal experimental results. The yield of the tomato substrate group was only comparable to that of the mushroom residue group. Mushroom residues have loose and porous structures. It was found that there were abundant nutrients in mushroom residues, such as crude proteins, crude fat, crude fibre, crude polysaccharides, trace elements, vitamins, minerals, and other nutrients essential for the growth and reproduction of edible mushrooms [8]. In this study, the elements in soil with mushroom residue increased the most significantly compared to other soil samples. Similarly, the yield was the highest, indicating that these elements were correlated with yield to some extent. However, the calcium ion content declined, which might be related to an imbalance of calcium ions due to the sharp increase in phosphorus and potassium. Adding coconut shells also led to substantial changes in physiochemical properties. Coconut shells are rich in cellulose, lignin, and nutrients (e.g., Ca^2+^). The Ca^2+^ concentration increased the most after the addition of coconut shells. The gradual degradation of fibre in coconut shells can provide carbon sources to microorganisms like Proteobacteria and Gemmatimonadetes to accelerate the degradation of substrates and the release of mineral elements [15]. It is important to note that the ammonia nitrogen content of different groups was lower than that of the control group, while the nitrate nitrogen content was higher. This is essentially the collaborative result of multiple mechanisms of action, such as the accelerated transformation of nitrogen forms, strengthening the nitrification and mineralization of organic matter after the addition of organic materials. Hence, it is necessary to further explore the relationship between physiochemical factors and key microorganisms.

### 4.3. Effects of Agricultural Waste on Bacteria in Cultivation Soil and Correlation Analysis of Physiochemical Factors

This study demonstrated that adding agricultural waste can influence bacterial structure and diversity in soil. On a phylum level, adding agricultural waste significantly influenced the abundance of beneficial bacteria Actinomycetota and Gemmatimonadota. In particular, the relative abundance of Actinomycetota and Gemmatimonadota increased significantly in the mushroom residue group. Actinomycetota plays a crucial role in organic matter degradation and nutrient cycling, and can effectively inhibit the transmission of plant pathogens, thus contributing positively to agricultural production [31]. Gemmatimonadota is vital in nitrogen and carbon cycling in soil and can decompose complicated organic matter and improve soil fertility [32]. On a genus level, *Devosia* from the tomato substrate group increased significantly. It has been shown that *Devosia* has nitrogen fixation and nitrification properties, playing an important role in agricultural productivity [33]. Previous studies have found that groups participating in nitrogen fixation and nitrification were part of the symbiotic network correlated with the high yield of *Morchella* spp. [13]. In this study, yields of experimental groups were significantly higher than that of the control group, and nitrogen content was higher, which might be related to an increase in *Devosia*. Additionally, beneficial bacteria like *Bacillaceae* increased significantly. The correlation analysis proved that TN, AP, AK, nitrate nitrogen, exchangeable magnesium, and available zinc were significantly positively correlated to Chloroflexota, Bacillota, and *Bacillaceae*, and significantly negatively correlated to Pseudomonadota and Bacteroidota. This suggests that adding different materials into soil can significantly affect bacterial communities by improving physiochemical properties.

### 4.4. Effects of Agricultural Waste on Fungi in Cultivation Soil and Correlation Analysis of Physiochemical Factors

It is reported that decreasing fungal diversity can decrease competition and make it favourable for *Morchella* spp. to grow [34]. The fungal communities in soil with different additives decreased to different extents. There were fewer fungal communities in soil with mushroom residues, indicating that adding agricultural waste can decrease competition and facilitate growth of *Morchella* spp. to some extent. The fungal community structures in soil with different additives were significantly different, but Ascomycota and Mortierellomycota still occupied dominant roles. There were common fungal phyla in soil after agricultural waste was added, but the relative abundance of *Mortierella* and *Trichoderma* decreased to some extent. During cultivation of *Morchella* spp., the common canker and white mould disease are the major fungal diseases. Research by Chen Cheng et al. on *Morchella* stem rot showed that the rhizosphere soil microbial community structure of the diseased fruiting body changed, and the dominant fungi included *Mortierella*. Given that *Mortierella* spp. are mostly saprophytic organisms, their quantitative dominance may be involved in the decay process of plant tissues, which is closely related to the occurrence of “rot stem” symptoms [35]. Li Xuesong et al. found that *Trichoderma* is the dominant microbial community in soil, which is a group related to large-sized fungal pathogenic bacteria [36]. According to the correlation analysis, physiochemical factors had some correlations to various key fungi in soil with different agricultural waste, but the correlation degrees were generally lower than those on a bacterial level. This proved that the addition of agricultural waste mainly influenced bacterial communities by changing physiochemical properties and affecting fungal communities to some extent, such as decreasing the content of pathomycete. They had important significance to the growth of *Morchella* spp. In this experiment, there were few types of pathomycetes in the different groups, and most common pathomycetes, like *Lecanicillium* [37], *Diplo9spora longispora* [38], and *Paecilomyces penicillatus* [39], were not detected. The prediction of fungal functions also showed that saprophytic nutritional fungi took the dominant role in all groups, while the proportion of pathological nutritional fungi was relatively low. The growth environment in this experiment was well controlled, and all devices were sterilized strictly before the experiment. Finally, the fruiting bodies of *Morchella* spp. were healthy. The analysis results and experimental results corroborated each other.

### 4.5. Effects of Soil Physiochemical Properties and Key Microorganisms on the Yield and Quality of Morchella spp.

Previous studies have demonstrated that fruiting body yield and quality of *Morchella* spp. are synchronized with organic matter, AK, and available nitrogen in the cultivation substrate, which are controlled by genetic and environmental factors [40]. Hence, factors related to the yield and quality of *Morchella* spp. were analysed in this study based on physical factors, chemical factors, and key microorganisms. Experimental results showed that the yield of *Morchella* spp. was significantly positively correlated with most physiochemical factors, including TN, AP, AK, nitrate nitrogen, exchangeable magnesium, and available zinc. Moreover, adding various organic matters in soil increased the yield of *Morchella* spp. by improving its physiochemical properties. Ammonia nitrogen had a significantly negative correlation with yield, showing a substantial correlation between nitrogen forms and yield. This agrees with previous studies [41]. The correlation analysis proved that adding different agricultural waste drives reconstruction of microbial community structures and functions by directly changing the physiochemical properties (e.g., nitrogen form, phosphorus, and potassium) of soil, influencing the yield and quality of *Morchella* spp. For example, a previous study found that groups participating in nitrogen fixation and nitrification are in the symbiotic network related to the high yield of *Morchella* spp. [13]. It has been shown that *Devosia* has nitrogen fixation and nitrification effects, and plays a crucial role in agricultural productivity [33]. This study proved that *Devosia* actually has a positive correlation with the yield of *Morchella* spp. Moreover, this study concluded that the yield of *Morchella* spp. significantly positively correlated with beneficial bacteria Actinomycetota, Bacillota, and *Bacillaceae*, but negatively correlated with harmful fungi *Mortierella* and *Trichoderma*. Hence, the microbial community structure can significantly affect the yield of *Morchella* spp. It is important to note that the relative abundance of *Bacillaceae* in the experimental groups was increased significantly compared to that of the control group. Xiao Qianming et al. invented a type of composite microbial agent and bio-organic fertilizer that can promote the growth of *Morchella* spp. This composite microbial agent is mainly composed of Bacillus, mixing several Bacillus strains. It can effectively promote the growth of *Morchella* spp., increase hypha activity and stability, and lower the manufacturing cost [42]. In this study, adding agricultural waste greatly increased the relative abundance of *Bacillaceae* to 4, 13, and 2 times that of the control group. Compared to tomato substrate and coconut shells, mushroom residues, an organic additive, changed the soil microbial structure more significantly and resulted in a higher yield. Edible mushroom residues are common agricultural waste and have been reused in soil improvement, animal feeds, edible mushroom crop cultivation, and raw materials for bioactivator extraction, but its utilization is still low [43]. This study revealed that nitrogen, phosphorus, potassium, and other nutrients are released during the degradation of mushroom residues, thus significantly improving soil fertility. Moreover, the relative abundance of *Bacillaceae* was significantly increased after mushroom residues were added, which shows significant positive correlations with the yield of *Morchella* spp. This study provides novel insights into the development of microbial agents and the significance to mushroom residue recycling.

## 5. Conclusions

In this study, the effects of agricultural waste (mushroom residues, waste tomato substrate, and coconut shells) on the yield and quality of *Morchella* spp., as well as the micro-ecology of cultivation soil, were discussed systematically. This study proves that the yield and quality of *Morchella* spp. increased significantly after adding agricultural waste. Agricultural waste can promote early fruiting and increase the yield of *Morchella* spp. Specifically, the mushroom residue group achieved the highest yield, which is related to nutrient enrichment in soil. With respect to quality, polysaccharides, crude protein, and potassium content increased significantly, while crude fat and crude fibre content decreased, which conforms to the modern demands of healthy food. Moreover, agricultural waste promoted the yield of *Morchella* spp. by improving the physical structure (e.g., lowering volume weight and increasing water holding capacity) and chemical properties (e.g., increasing nitrogen, phosphorus, and potassium, changing nitrogen forms, and increasing Mg+, Zn+, and other ions) of soil. The relative abundance of beneficial bacteria like Actinomycetota, Gemmatimonadota, and *Devosia* increased significantly, and their nitrogen fixation, nitrification, and organic matter decomposition directly determined yield growth. The relative abundance of pathomycete (e.g., *Mortierella* and *Trichoderma*) declined. Decreased fungal diversity relieved ecological competition and further optimized the growth environment for *Morchella* spp. Spent mushroom substrate has the dual advantages of nutrient release and microbial regulation, which is an ideal organic additive. Tomato substrate and coconut shell also showed good application potential. In conclusion, agricultural waste can significantly increase the yield and quality of *Morchella* spp. through a synergistic path of “optimization of physiochemical properties/reconstruction of microbial communities”. Future studies should focus on the targeted regulation of specific functional microorganisms (e.g., nitrogen-fixing bacteria and Bacillus), further decreasing ecological risks, and promoting the sustainable development of *Morchella* spp. cultivation by combining waste preprocessing technologies (e.g., leaching and fermentation).

## Figures and Tables

**Figure 1 jof-11-00703-f001:**
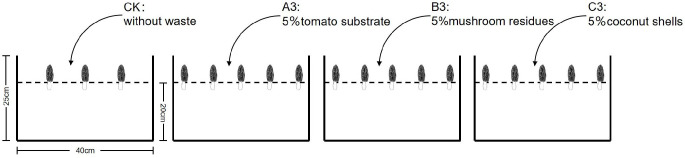
Culture diagram of *Morchella* spp.

**Figure 2 jof-11-00703-f002:**
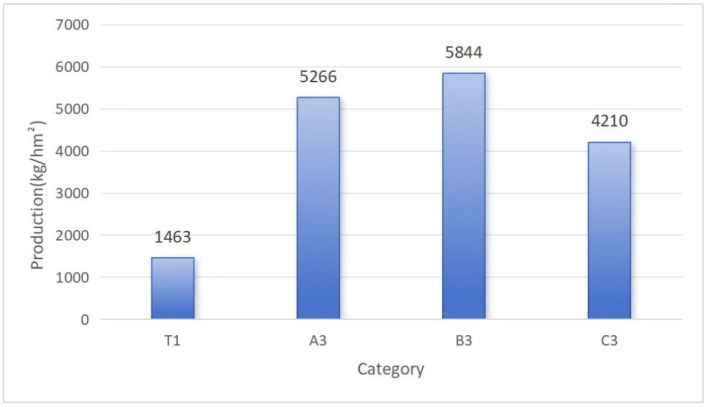
*Morchella* spp. yield under different substrates.

**Figure 3 jof-11-00703-f003:**
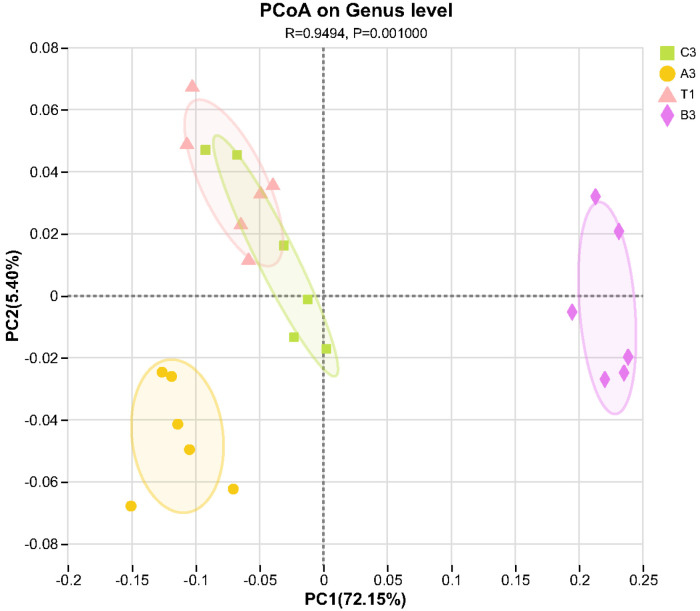
PCoA analysis on bacterial community structures in cultivation soils with different additives.

**Figure 4 jof-11-00703-f004:**
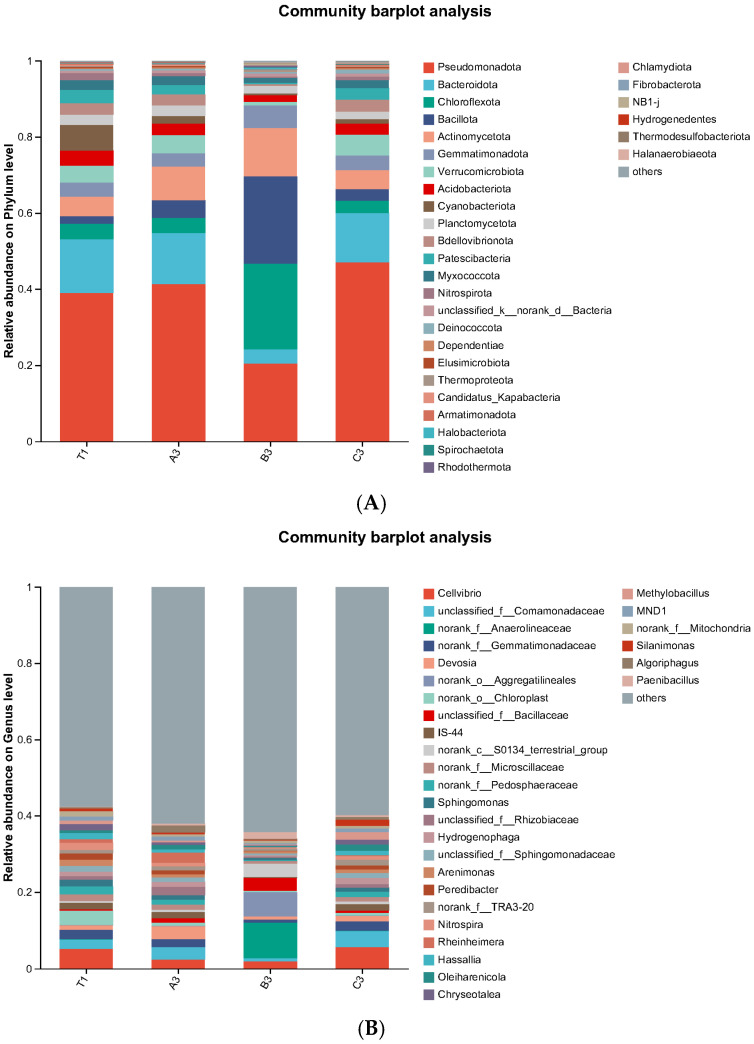
(**A**) Bacterial community distribution in cultivation soils with different additives on the phylum level. (**B**) Bacterial community distribution in cultivation soils with different additives on the genus level.

**Figure 5 jof-11-00703-f005:**
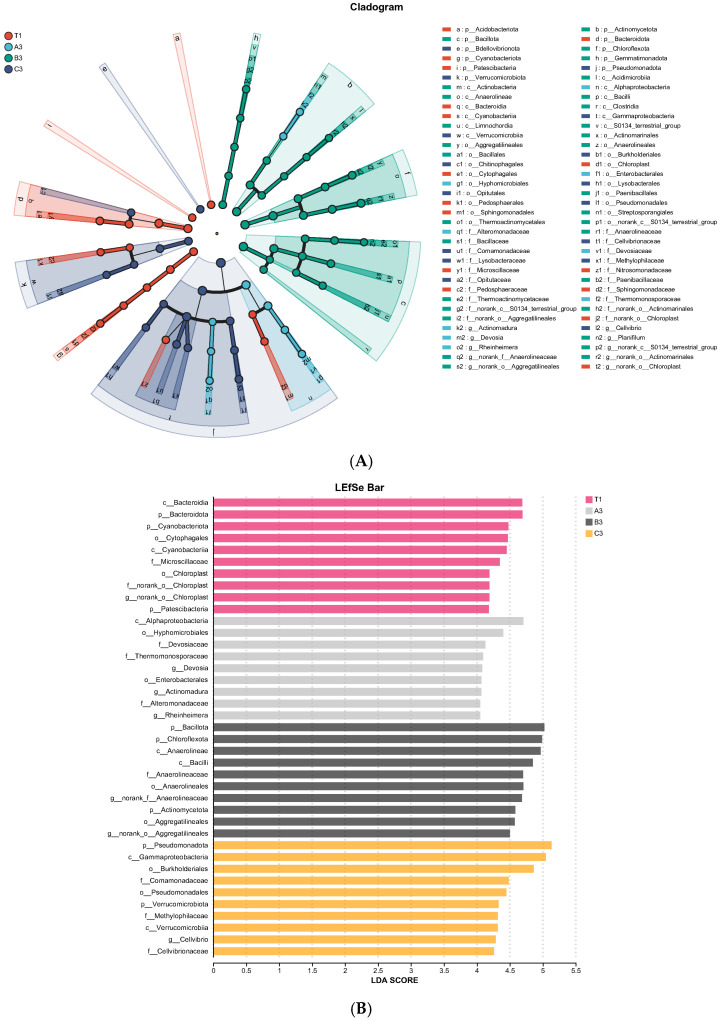
(**A**) LEfSe ring-shaped tree chart of bacterial communities in cultivation soils. (**B**) *t*-test analysis chart of bacterial communities in cultivation soils.

**Figure 6 jof-11-00703-f006:**
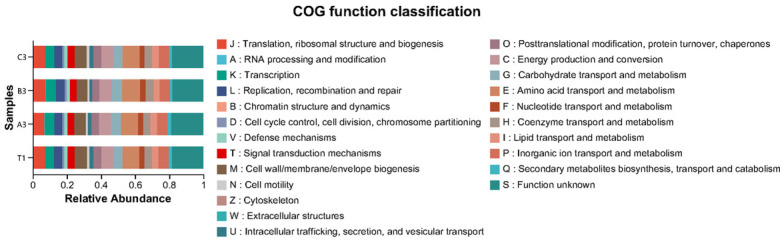
Prediction of bacterial functions in cultivation soil based on PICRUSt2 annotations.

**Figure 7 jof-11-00703-f007:**
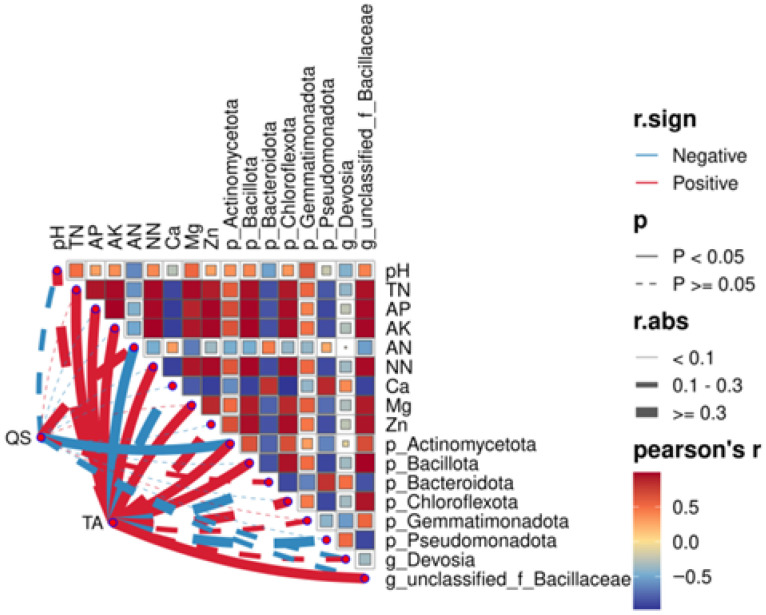
Correlation analysis based on bacterial community.

**Figure 8 jof-11-00703-f008:**
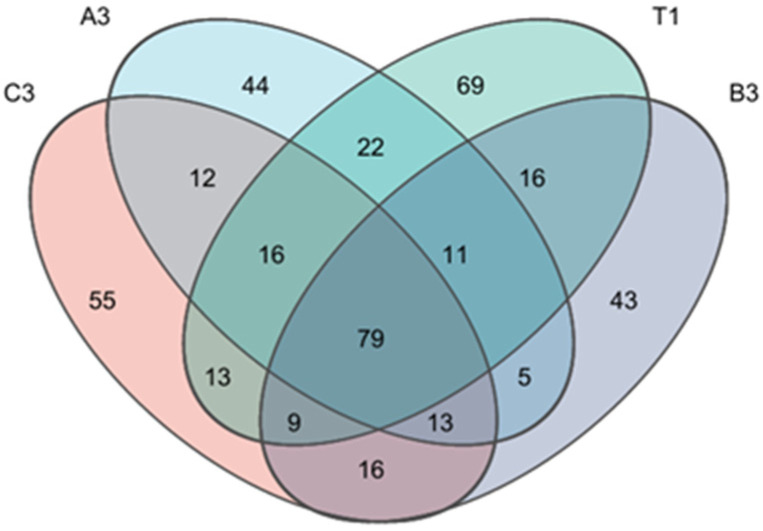
Venn diagram of abundance of fungi communities in cultivation soils.

**Figure 9 jof-11-00703-f009:**
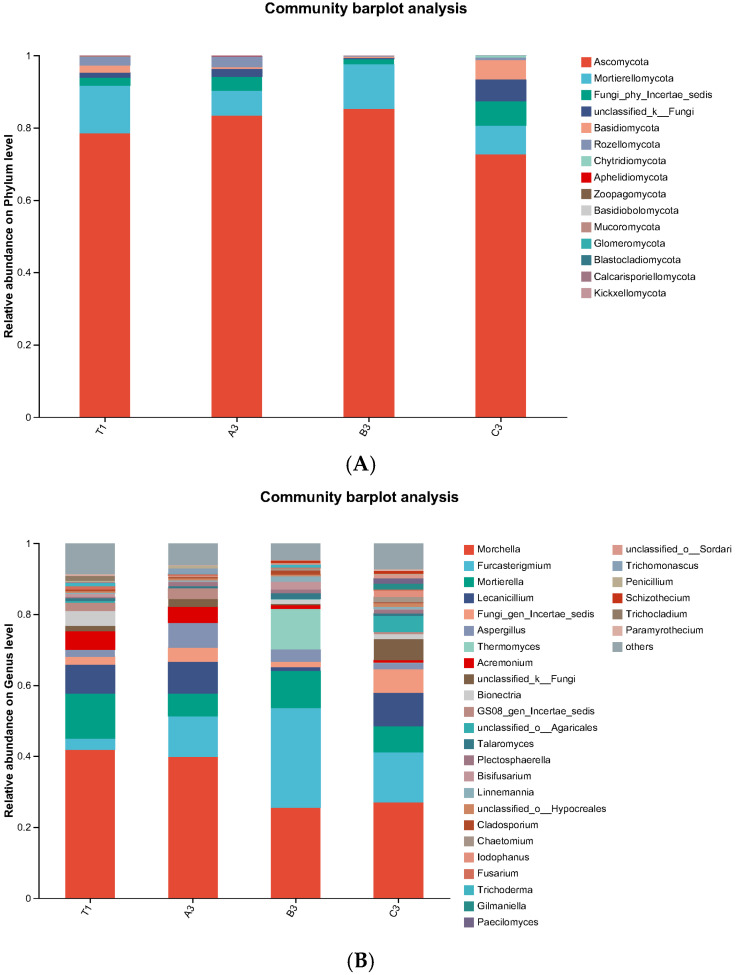
(**A**) Fungi community distribution in soil with different additives at a phylum level. (**B**) Fungi community distribution in soil with different additives at a genus level.

**Figure 10 jof-11-00703-f010:**
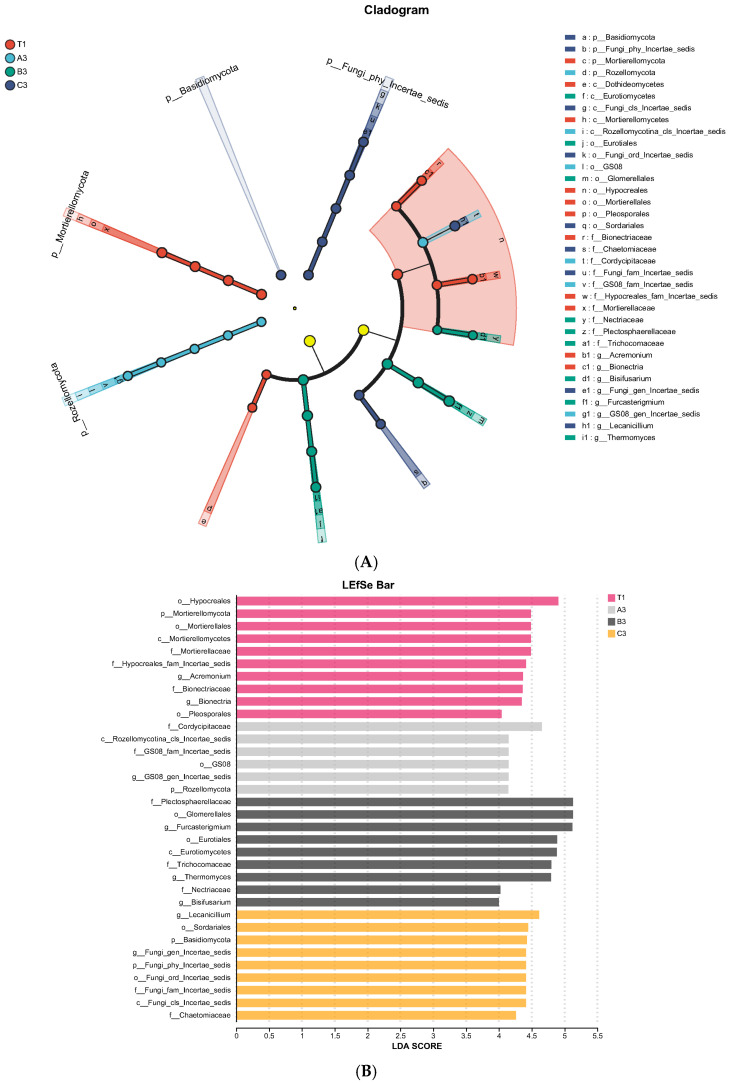
(**A**) LEfSe ring-shaped tree chart of fungal communities in cultivation soils. (**B**) *t*-test analysis chart of fungal communities in cultivation soils.

**Figure 11 jof-11-00703-f011:**
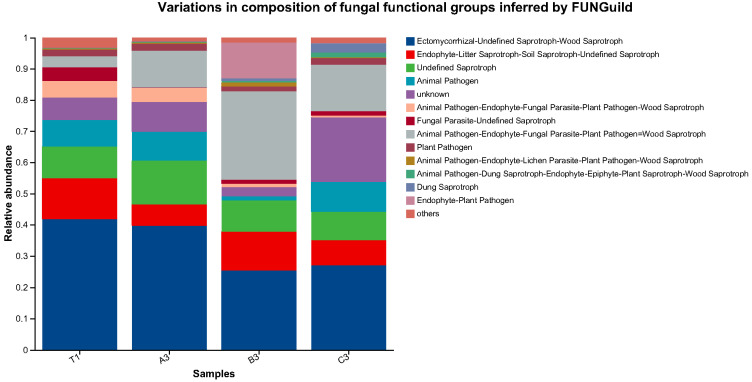
Prediction of potential functions of fungi in cultivation soils.

**Figure 12 jof-11-00703-f012:**
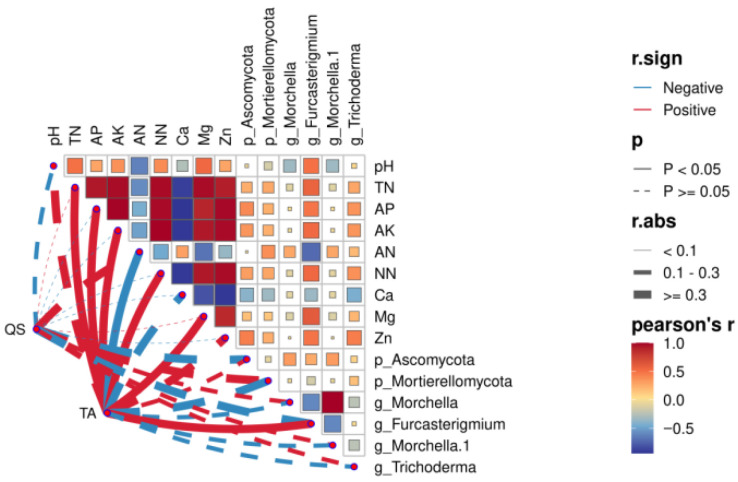
Correlation analysis based on fungal communities.

**Table 1 jof-11-00703-t001:** Primer sequences to determine microbial diversity.

Sequencing Region	Primer Name	Primer Sequence (5′→3′)
ITS1F_ITS2R	ITS1F	CTTGGTCATTTAGAGGAAGTAA
	ITS2R	GCTGCGTTCTTCATCGATGC
ArBa515F_806R	ArBa515F	GTGCCAGCMGCCGCGGTAA
	806R	GGACTACHVGGGTWTCTAAT

**Table 2 jof-11-00703-t002:** Nutrient content of fruiting bodies under different substrates.

Groups	Polysaccharides	Crude Protein	Total Potassium Ion	Crude Fat	Crude Fibre	Water Content
	g/kg	g/100 g	g/100 g	g/100 g	g/100 g	g/100 g
T1	30.08 ± 0.62 c	43.19 ± 0.04 a	3.43 ± 0.03 d	2.44 ± 0.07 a	20.24 ± 0.94 a	89.17 ± 0.17 c
A3	32.83 ± 1.18 b	42.21 ± 0.07 b	3.62 ± 0.02 b	2.13 ± 0.04 c	17.00 ± 0.76 b	90.38 ± 0.26 b
B3	43.62 ± 0.65 a	43.23 ± 0.04 a	3.54 ± 0.01 c	2.26 ± 0.06 b	16.34 ± 0.32 b	87.66 ± 0.23 d
C3	33.55 ± 1.08 b	41.52 ± 0.04 c	3.90 ± 0.01 a	2.12 ± 0.07 c	17.07 ± 1.27 b	91.59 ± 0.06 a

Different lowercase letters in the same row indicate significant difference (*p* < 0.05).

**Table 3 jof-11-00703-t003:** Test results of physiochemical properties of soil samples with different additives.

Groups	Water Content	pH	TN	AP	AK	Ammonium Nitrogen	Nitrate Nitrogen	Exchangeable Calcium	Exchangeable Magnesium	Available Zinc
	%	Dimensionless	g/kg	mg/kg	mg/kg	mg/kg	mg/kg	Cmol/kg	Cmol/kg	mg/kg
T1	24.65 ± 0.07 d	8.12 ± 0.07 b	0.68 ± 0.02 d	17.28 ± 0.65 c	254.52 ± 3.45 d	13.62 ± 0.16 a	10.60 ± 0.64 c	9.48 ± 0.22 a	1.04 ± 0.01 d	1.67 ± 0.05 b
A3	31.85 ± 0.03 a	8.16 ± 0.07 b	0.70 ± 0.02 c	20.04 ± 0.40 b	323.01 ± 2.86 b	7.29 ± 0.14 b	11.04 ± 0.85 bc	9.51 ± 0.21 a	1.12 ± 0.04 c	1.68 ± 0.03 b
B3	27.31 ± 0.19 c	8.27 ± 0.04 a	1.03 ± 0.01 a	38.99 ± 0.69 a	960.46 ± 8.67 a	5.29 ± 0.12 c	36.49 ± 3.10 a	8.63 ± 0.16 b	1.53 ± 0.03 a	2.30 ± 0.07 a
C3	29.46 ± 0.10 b	8.30 ± 0.04 a	0.77 ± 0.02 b	16.65 ± 0.40 c	316.07 ± 2.62 c	5.13 ± 0.16 d	13.00 ± 0.77 b	9.67 ± 0.28 a	1.23 ± 0.04 b	1.59 ± 0.02 c

Different lowercase letters in the same row indicate significant difference (*p* < 0.05).

## Data Availability

The original contributions presented in this study are included in the article. Further inquiries can be directed to the corresponding authors.

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
