# Peer review of "Improving the Yield and Quality of Morchella spp. Using Agricultural Waste"

_jof, 2025, doi:10.3390/jof11100703_

Round 1
Reviewer 1 Report
This study evaluates tomato substrate, mushroom residues, and coconut shells as soil amendments to improve Morchella spp. cultivation. Results show substantial yield increases, better nutritional quality, enhanced soil properties, enrichment of beneficial microbes, and suppression of pathogenic fungi. The work provides a clear mechanism linking soil improvements to yield gains and represents a valuable contribution to sustainable mushroom production.
Minor Revisions and Suggestions
- Italicize all genus/species (Morchella, Mortierella, Trichoderma, Devosia).
- Unify “total potassium” notation (TK vs TP); pick one and update text, tables, and figures.
- Specify tests (e.g., ANOVA + Tukey), and clarify how biological replicates (n=6) were treated (analyzed individually vs. pooled).
- State software/versions (QIIME2/DADA2/Deblur), databases, and parameters for PICRUSt2/FUNGuild.
- Clarify whether temperature/air/soil humidity were monitored and maintained throughout, not only at planting.
- Results §2.3.5 report yield negatively correlated with nitrate-N; Discussion §3.5 states yield positively correlated with nitrate-N and negatively with ammonium-N. Reconcile across text, figures, and supplements.
- Mortierella often includes saprotrophs; refine claims about “harmful/pathogenic” lineages by specifying OTUs/taxa implicated in Morchella systems and provide supporting citations.
- For coconut shells → K enrichment, report measured soil extractable K (not just fruiting-body K).
- For mushroom residues → higher polysaccharides, connect to measured labile C pools or enzyme activities if available (or note as inference).
- Provide pre-/post-leaching EC (mS·cm⁻¹) and pH, plus thresholds defining “normal ranges.”
Questions for Authors
- What were the C:N, moisture, and maturity/composting status for each additive? Could these explain treatment differences?
- What were the pre/post 24-h leaching EC and pH of the tomato substrate, and what thresholds define “within normal ranges”?
- Have you tested blends (e.g., mushroom residues + coconut shells) or higher inclusion rates (10–20%) to explore trade-offs among water retention, nutrient release, and microbiome restructuring?
- Do you have strain-level evidence (ASVs/qPCR) pointing to specific Bacillus spp. analogous to commercial inoculants, or plans to isolate/test strains from high-yield plots?
- Can you provide a simple cost–benefit estimate (input costs, pretreatment labor, yield gain per ha) relative to conventional practice?
- What multi-season effects on soil health/microbiome do you anticipate, and are follow-up trials planned?
This study evaluates tomato substrate, mushroom residues, and coconut shells as soil amendments to improve Morchella spp. cultivation. Results show substantial yield increases, better nutritional quality, enhanced soil properties, enrichment of beneficial microbes, and suppression of pathogenic fungi. The work provides a clear mechanism linking soil improvements to yield gains and represents a valuable contribution to sustainable mushroom production.
Minor Revisions and Suggestions
- Italicize all genus/species (Morchella, Mortierella, Trichoderma, Devosia).
- Unify “total potassium” notation (TK vs TP); pick one and update text, tables, and figures.
- Specify tests (e.g., ANOVA + Tukey), and clarify how biological replicates (n=6) were treated (analyzed individually vs. pooled).
- State software/versions (QIIME2/DADA2/Deblur), databases, and parameters for PICRUSt2/FUNGuild.
- Clarify whether temperature/air/soil humidity were monitored and maintained throughout, not only at planting.
- Results §2.3.5 report yield negatively correlated with nitrate-N; Discussion §3.5 states yield positively correlated with nitrate-N and negatively with ammonium-N. Reconcile across text, figures, and supplements.
- Mortierella often includes saprotrophs; refine claims about “harmful/pathogenic” lineages by specifying OTUs/taxa implicated in Morchella systems and provide supporting citations.
- For coconut shells → K enrichment, report measured soil extractable K (not just fruiting-body K).
- For mushroom residues → higher polysaccharides, connect to measured labile C pools or enzyme activities if available (or note as inference).
- Provide pre-/post-leaching EC (mS·cm⁻¹) and pH, plus thresholds defining “normal ranges.”
Questions for Authors
- What were the C:N, moisture, and maturity/composting status for each additive? Could these explain treatment differences?
- What were the pre/post 24-h leaching EC and pH of the tomato substrate, and what thresholds define “within normal ranges”?
- Have you tested blends (e.g., mushroom residues + coconut shells) or higher inclusion rates (10–20%) to explore trade-offs among water retention, nutrient release, and microbiome restructuring?
- Do you have strain-level evidence (ASVs/qPCR) pointing to specific Bacillus spp. analogous to commercial inoculants, or plans to isolate/test strains from high-yield plots?
- Can you provide a simple cost–benefit estimate (input costs, pretreatment labor, yield gain per ha) relative to conventional practice?
- What multi-season effects on soil health/microbiome do you anticipate, and are follow-up trials planned?
Reviewer 2 Report
Dear authors, thank you for providing this article. It examines the influence of soil composition on the cultivation of the edible mushroom Morchella spp. The authors studied the addition of tomato substrate, mushroom residues, and coconut shells to the soil on mushroom growth. The authors attribute improved mushroom growth in soil with the addition of these substances to changes in nitrogen metabolism and an increase in the number of beneficial soil bacteria. The authors found mushroom residues to be the most effective fertilizer for mushroom cultivation. The article is comprehensive, containing many different modern and classical methods. The discussion is logical. I believe the article would be even more interesting if, in addition to the methods used for introducing organic amendments during mushroom cultivation, the authors examined the use of these amendments in combination and all three together. If such results are found, they can be added.
After minor revision, the article may be published in the journal "JoF."
Respectfully yours, reviewer.
September 20, 2025
The article needs proofreading for typos (for example, a period is missing at the end of the sentence – line 64), and formatting according to the journal's guidelines (see the bibliography).
I would like the authors to add a section on statistical analysis of the results to the Methodological Section of the paper, and to indicate in Table 2 what data are presented – is it the mean value and the error?
The text in Figures 4, 5, 6, 9, 10, and 11 is unreadable; it is too small.
The conclusion is too extensive. I recommend shortening it.
Author Response
Comment 1:The article needs proofreading for typos (for example, a period is missing at the end of the sentence – line 64), and formatting according to the journal's guidelines (see the bibliography).
Reply 1:Thank you for pointing out this point, I agree with this opinion. I have proofread typos, and adjust the format fonts, etc.
Comment 2:I would like the authors to add a section on statistical analysis of the results to the Methodological Section of the paper, and to indicate in Table 2 what data are presented – is it the mean value and the error?
Reply 2:Thank you for pointing out this point, I agree with this opinion. The method part of the paper has been added, and Table 2 has been explained.
Comment 3:The text in Figures 4, 5, 6, 9, 10, and 11 is unreadable; it is too small.
Reply 3:Thank you for pointing out this point, I agree with this opinion. I ' ve replaced the unclear pictures with texts.
Comment 4:The conclusion is too extensive. I recommend shortening it.
Reply 4:Thank you for pointing out this point, I agree with this opinion. I have revised it at the conclusion